# Non-Equilibrium Long-Wave Infrared HgCdTe Photodiodes: How the Exclusion and Extraction Junctions Work Separately

**DOI:** 10.3390/ma17112551

**Published:** 2024-05-25

**Authors:** Małgorzata Kopytko, Kinga Majkowycz, Jan Sobieski, Tetiana Manyk, Waldemar Gawron

**Affiliations:** 1Institute of Applied Physics, Military University of Technology, 2 Kaliskiego St., 00-908 Warsaw, Poland; malgorzata.kopytko@wat.edu.pl (M.K.); jan.sobieski@wat.edu.pl (J.S.); tetjana.manyk@wat.edu.pl (T.M.); waldemar.gawron@wat.edu.pl (W.G.); 2Vigo Photonics S.A., 129/133 Poznańska St., 05-850 Ożarów Mazowiecki, Poland

**Keywords:** exclusion and extraction effect, Auger suppression, non-equilibrium conditions, infrared detectors, HgCdTe photodiode

## Abstract

The cooling requirement for long-wave infrared detectors still creates significant limitations to their functionality. The phenomenon of minority-carrier exclusion and extraction in narrow-gap semiconductors has been intensively studied for over three decades and used to increase the operating temperatures of devices. Decreasing free carrier concentrations below equilibrium values by a stationary non-equilibrium depletion of the device absorber leads to a suppression of Auger generation. In this paper, we focus on analyzing exclusion and extraction effects separately, based on experimental and theoretical results for a HgCdTe photodiode. To carry out an experiment, the n^+^-P^+^-π-N^+^ heterostructure was grown by metal organic chemical vapor deposition on CdTe-buffered GaAs substrate. In order to separate the extraction and exclusive junctions, three different devices were evaluated: (1) a detector etched through the entire n^+^-P^+^-π-N^+^ heterostructure, (2) a detector made of the P^+^-π photoconductive junction and (3) a detector made of the π-N^+^ photodiode junction. For each device, the dark current density–voltage characteristics were measured at a high-temperature range, from 195 K to 300 K. Next, the carrier concentration distribution across the entire heterostructure and individual junctions was calculated using the APSYS simulation program. It was shown that when the n^+^-P^+^-π-N^+^ photodiode is reverse biased, the electron concentration in the π absorber drops below its thermal equilibrium value, due to the exclusion effect at the P^+^-π junction and the extraction effect at the π-N^+^ junction. To maintain the charge neutrality, the hole concentration is also reduced below the equilibrium value and reaches the absorber doping level (*N_A_*), leading to the Auger generation rate’s reduction by a factor of 2*n_i_*/*N_A_*, where *n_i_* is the intrinsic carrier concentration. Our experiment conducted for three separate detectors showed that the exclusion P^+^-π photoconductive junction has the most significant effect on the Auger suppression—the majority of the hole concentration drops to the doping level not only at the P^+^-π interface but also deep inside the π absorber.

## 1. Introduction

Mercury cadmium telluride (Hg_1−*x*_Cd*_x_*Te, MCT) is still one of the key materials used in infrared (IR) technology of high-operating-temperature (HOT) devices [1]. The specific advantages of HgCdTe are that they have a direct energy gap, are tunable in a wide energy range and make it possible to obtain both low and high carrier concentrations; these features enable the design of devices based on heterostructures. The use of a double-heterostructure (DH) photodiode proposed by British scientists [2,3,4,5], in which the active region is sandwiched between wide gap layers (P^+^-π-N^+^ or P^+^-ν-N^+^ design, where the capital letter denotes a wider-gap semiconductor, the upper index “+” denotes heavy doping and π and ν denote the low-doped p or n region, respectively) has created the possibility of increasing the operating temperature of IR detectors without deteriorating their parameters. The operation of HOT IR detectors has become an almost universal goal in the last three decades, with particular importance in the long-wave infrared (LWIR) range. In narrow-gap HgCdTe (with a bandgap energy of about 0.1–0.15 eV), a high intrinsic carrier concentration at ambient temperature is the source of leakage currents resulting from thermal generation processes. In order to suppress generation–recombination (GR) processes, primarily the Auger mechanism, non-equilibrium operating conditions are used. In the P^+^-π-N^+^ photodiode illustrated in Figure 1, the intrinsically generated electrons are extracted from the absorber region by a positive electrode connected to the N layer. Electrons are also excluded from the absorber region near the P-π junction because they cannot be injected from the P layer. As a consequence of the decrease in electron concentration (*n*) to almost zero, the hole concentration (*p*) decreases to the acceptor doping level (*N_A_*), leading to a reduction in the Auger generation rate, and consequently dark current, by a factor of 2*n_i_*/*N_A_*, where *n_i_* is the intrinsic carrier concentration, with an improvement in detectivity by (2*n_i_*/*N_A_*)^1/2^ [6].

We note that there are extensive experimental and theoretical analyses in the literature of the extraction and exclusion phenomenon [7,8,9,10,11,12,13]. Here, we focus on analyzing these two effects separately based on experimental and theoretical results for an LWIR HgCdTe photodiode. The n^+^-P^+^-π-N^+^ heterostructure was grown by metal organic chemical vapor deposition (MOCVD) on CdTe-buffered GaAs substrate. In order to separate the extraction and exclusive junctions, three different experiments were performed: the measurement of the dark current–voltage (*J*–*V*) characteristics for the entire n^+^-P^+^-π-N^+^ device, as well as for the P^+^-π photoconductive and the π-N^+^ photodiode junctions. Finally, a theoretical analysis was performed using the commercially available APSYS platform (Crosslight Software Inc., Burnaby, BC, Canada).

## 2. Experiment

The (111)B HgCdTe epitaxial layer was grown in a horizontal, near-atmospheric-pressure Aixtron AIX-200 MOCVD system on a 2-inch (100) GaAs substrate. The CdTe buffer layer was used in order to compensate for the lattice mismatch between the HgCdTe epilayer and the GaAs substrate. The growth was carried out at a temperature of about 350 °C and a mercury zone of 210 °C using the interdiffused multilayer process (IMP) [14]. More comprehensive details of the growth experiments performed in our laboratory are presented in [15,16].

The analyzed device is a four-layer n^+^-P^+^-π-N^+^ photodiode with a ~5 µm thick p-type arsenic-doped (*N_A_* = 3 × 10^15^ cm^–3^) narrow-gap absorber (Cd molar fraction of *x* = 0.225), sandwiched between wide-gap P^+^ (*x* = 0.396) and N^+^ (*x* = 0.44, *N_D_* = 2 × 10^17^ cm^–3^) layers. The P^+^ layer was heavily doped with arsenic (*N_A_* = 4 × 10^17^ cm^–3^), while the N^+^ layer was doped with iodine at the level of *N_D_* = 2 × 10^17^ cm^–3^. The cross-section of the analyzed n^+^-P^+^-π-N^+^ HgCdTe photodiode is presented in Figure 2. The device is essentially a P^+^-π-N^+^ photodiode. The top n^+^ layer was deposited on the P^+^ layer to provide low-resistance ohmic contact. The transient-layer (T) parameters were conditioned by interdiffusion processes during the MOCVD growth at 350 °C.

Square 400 μm × 400 μm mesa photodetectors were defined using standard UV photolithography and wet chemical etching using a Br:HBr (1:100) solution diluted in deionized water (50:50:1 Br:HBr:H_2_O). Three different experiments were performed to separate the extraction and exclusive junctions. 

In the first experiment (E1), deep etching of the mesa detector through the entire heterostructure was performed and metallic contacts were made to the top n^+^ and the bottom N^+^ layers—measurement was performed for the entire n^+^-P^+^-π-N^+^ photodiode. 

In the second experiment (E2), the mesa detector was etched into the absorber and metallic contacts were made to the top n+ contact and the π-absorber—the measurement was performed for the n^+^-P^+^-π device consisting of the P^+^-π photoconductive junction. 

In the third experiment (E3), the upper layers were etched off to expose the absorber; then, the mesa detector was etched into the bottom contact, and metallic contacts were made to the π-absorber and the bottom N^+^ layer—the measurement was performed for the π-N^+^ photodiode junction.

Spectral response (SR) measurements were performed in the back-side illuminated n^+^-P^+^-π-N^+^ photodiode using a Spectrum 2000-type FTIR Perkin Elmer spectrometer. The absolute current responsivity (*R_i_*) values, in [A/W], were obtained by a measuring station composed of a lock-in analyzer, a monochromator and a chopped 1000 K blackbody. The current density–voltage (*J–V*) characteristics were collected using the Semetrol system in the bias voltage range from 0.2 V to –0.8 V. During the measurements, the detector was mounted in a helium closed-cycle cryostat which ensured temperature stabilization within the range of 50 K to 300 K. The paper presents measurement results in the range of temperatures reached by thermoelectric coolers commercially used for HOT detectors.

Theoretical modeling was performed using the commercially available APSYS platform. The detectors performances were made taking into account a wide spectrum of GR mechanisms: Auger, Shockley*–*Reed*–*Hall (SRH) and tunneling processes. Calculations of carrier concentration distribution were made across the entire n^+^-P^+^-π-N^+^ heterostructure and for individual n^+^-P^+^-π and π-N^+^ junctions. In the calculations, all contacts were assumed to be ohmic.

## 3. Experimental Results

Figure 3 shows the spectral *R_i_* characteristics of the LWIR n^+^-P^+^-π-N^+^ HgCdTe photodiode measured at four temperatures. The value of the maximum *R_i_* does not change much with temperature and is at the level of ~2.4 A/W. The cut-off wavelength is consistent with the theoretical relationship for HgCdTe [17] with *x* = 0.225 and decreases from ~8 μm to ~6.5 μm in the temperature range from 195 K to 300 K. It should be noted that the energy gap of the narrow-gap HgCdTe, unlike III–V semiconductors [18], widens with increasing temperature.

In a given temperature range, the dark *J–V* characteristics were measured and are plotted in Figure 4. Their shape is determined by the effect of carrier exclusion, and extraction occurred under non-equilibrium conditions. Let us analyze the plot shape of a reverse biased photodiode at 300 K. The reverse voltage range of can be divided into three areas: (i) in the bias range from 0 V to –0.06 V—the dark current density increases to the value of 4.63 A/cm^2^. It should be noted that this is not the maximum current density (*J_dmax_*) value that would result from intrinsically generated carriers within the absorber. (ii) From –0.06 V to –0.25 V—a further increase in the reverse bias voltage causes the dark current density to drop to the minimum value (*J_dmin_*) of 1.23 A/cm^2^, due to the exclusion and extraction of carriers from the absorber. (*J_dmin_*) is determined by the absorber doping level. (iii) Above –0.25 V—a further increase in the reverse bias voltage causes a gradual increase in the dark current density. In an ideal photodiode, this current should be saturated. The observed current increase is associated with parasitic leakage mechanisms, such as the GR current through SRH centers within the semiconductor bandgap, and/or tunnel currents (band to band and/or via SRH centers), and/or surface leakage current, especially dominant in the p-type HgCdTe [19].

As can be seen in Figure 4, the GR mechanisms’ suppression due to the non-equilibrium conditions is more efficient at high temperatures. This is due to the larger *n_i_*/*N_A_* ratio at higher temperatures.

Assume that in a high-quality HgCdTe photodiode, the defect-related SRH generation [20,21] is not an issue, and the radiative one is also negligible due to the photon reabsorption [22]. Then, the thermal generation rate in the π absorber (*p* >> *n*) is limited by the Auger 7 mechanism, giving the dark current density expressed as [23]
(1)$$J_{d}=\frac{qt(p+n)}{2\tau_{Ai7}},$$
where *q* is the electric charge, *t* is the absorber thickness and *τ_Ai7_* is the Auger 7 lifetime in an intrinsic material [24].

At high temperatures, where *p* = *n* = *ni*, the generation rate maintaining a thermal equilibrium within the field-free absorber gives the *J_dmax_* value of
(2)$$J_{dmax}=\frac{qtn_{i}}{\tau_{Ai7}}$$

So, the thermal equilibrium dark current density is determined by the intrinsically generated carriers in the absorber.

Under the non-equilibrium mode, when the electron concentration decreases to almost zero, *n* = 0, and the hole concentration decreases to the acceptor doping level, *p* = *N_A_*, the dark current density drops to the *J_dmin_* value of
(3)$$J_{dmin}=\frac{qtN_{A}}{2\tau_{Ai7}}$$

In other words, the dark current density decreases by a factor of 2*n_i_*/*N_A_*.

Figure 5 shows the intrinsic carrier concentration (*ni*) versus cut-off wavelength calculated at four temperatures. The absorber doping level of the analyzed photodiode and the intrinsic concentration at a given temperature is also marked. At 300 K, the As concentration in the absorber region is almost an order of magnitude lower than the intrinsic carrier concentration. At 195 K, the absorber doping level is only slightly lower than the intrinsic concentration; therefore, the difference between the *J_dmin_* and *J_dmax_* values is also insignificant.

The anode IR detector parameter limited by the statistical character of GR processes in the material is the normalized detectivity, *D**. For a device operating with a reverse bias, *D** can be given by [25]
(4)$$D^{*}=\frac{R_{i}}{{\left(2qJ_{d}\right)}^{1/2}}$$
expressed in [cmHz^1/2^/W], otherwise called [Jones]. The effect of exclusion and extraction under non-equilibrium conditions therefore leads to an improvement in detectivity by (2*n_i_*/*N_A_*)^1/2^.

Figure 6 shows the spectral *D** characteristics of the LWIR n^+^-P^+^-π-N^+^ HgCdTe photodiode determined on the basis of the spectral *R_i_* characteristics and *J_dmin_* values appropriate for a given temperature. The uncooled (300 K) non-equilibrium photodiode reaches a detectivity level of 4 × 10^9^ cmHz^1/2^/W. A similar photodiode limited by intrinsically generated carriers would have a detection range not exceeding 2 × 10^9^ cmHz^1/2^/W. However, the Auger-suppressed device is still by one order of magnitude below the background limit of performance.

## 4. Calculation Results

Figure 7 shows a calculated carrier concentration distribution across the P^+^-π-N^+^ heterostructure. Calculations were performed at 230 K and different reverse bias voltages. Solid lines represent equilibrium conditions for a zero-bias voltage, while dotted/dashed lines represent non-equilibrium carrier concentrations calculated at different reverse bias voltages. Under equilibrium conditions, the hole concentration in the absorber is determined by the intrinsic one, which is at the level of *n_i_* = 8 × 10^15^ cm^–3^. With the increasing reverse bias voltage, the intrinsically generated electrons are extracted from the absorber region by the π-N^+^ junction, and at the same time are excluded from the absorber near the P-π junction (Figure 7a). As a consequence, the hole concentration decreases to the acceptor doping level (*N_A_* = 3 × 10^15^ cm^–3^) already for a voltage of –0.1 V (Figure 7b).

Figure 8 shows a calculated carrier concentration distribution across the P^+^-π-N^+^ heterostructure at 300 K. At room temperature, the intrinsic carrier concentration is at the level of *n_i_* = 2 × 10^16^ cm^–3^. Here, the drop in hole concentration is more pronounced, but only at a voltage of –0.2 V. This voltage coincides with the point of the drop in the dark current, which is visible in the *J–V* characteristics (see Figure 4). Both the exclusion and extraction junctions in the P^+^-π-N^+^ photodiode ensure a uniform reduction in the hole non-equilibrium concentration to the doping level throughout the entire absorber thickness. Next, we will show how individual junctions work separately. The P^+^-π photoconductive junction on the left side of the chart is referred to as the exclusion junction, and the π-N^+^ photodiode junction on the right side of the chart is referred to as the extraction one.

The calculated electron and hole concentration distribution in three different experiments are presented in Figure 9 and Figure 10 for 230 K and 300 K, respectively. Under the reverse bias conditions (–0.2 V), the minority electrons are extracted from the π absorber near the π-N^+^ interface. In turn, the P^+^-π extraction junction does not allow electrons to be injected into the π absorber. As a result, the electron concentration drops below its thermal equilibrium value by three orders of magnitude on the P^+^-π interface and by just over an order of magnitude on the π-N^+^ interface at 230 K. To maintain the charge neutrality, the hole concentration is also reduced below the equilibrium value and reaches the doping level at the appropriate junctions. Furthermore, the penetration depth of the exclusive region is much greater than that of the extraction one. At 300 K, the electron concentration drops by about an order of magnitude at both the P^+^-π and π-N^+^ junctions.

Similarly to the entire n^+^-P^+^-π-N^+^ photodiode (see Figure 4), the exclusion and extraction effect are separately followed by Auger generation suppression. Figure 11 shows the dark *J–V* characteristics measured for three different devices (n^+^-P^+^-π-N^+^, P^+^-π and π-N^+^) at 230 K and 300 K. The decrease in dark current due to the exclusion effect at the P^+^-π junction (dotted red lines) is not as effective as for the π-N^+^ extraction junction (dashed green lines) and is visible only at room temperature. At 230 K, there is no drop in dark current in the P^+^-π detector, but the dark current increases linearly with reverse bias voltage.

## 5. Conclusions

Non-equilibrium-mode Auger suppression is a viable approach to drive the operating temperature of the IR photodetectors. The elimination of cooling requirements leads to significant cost reduction, logistical supply, and an increase in their application area. 

An experiment was performed to estimate the impact of the non-equilibrium exclusive and extraction effects on the generation of dark current in an LWIR n^+^-P^+^-π-N^+^ HgCdTe photodiode. Three different devices were evaluated, and for each device, the measured *J–V* characteristics showed dark current reduction due to Auger suppression. As the theoretical analysis shows, the stronger and more dominant effect is the exclusion at the P^+^-π junction. However, the most effective result is achieved when they are combined in a P^+^-π-N^+^ heterostructure photodiode. Then, the electron concentration in the entire π absorber drops below its thermal equilibrium value, due to the exclusion at the P^+^-π junction and the extraction at the π-N^+^ junction. The hole concentration is also reduced below the equilibrium value and reaches the absorber doping level, leading to dark current reduction by a factor of 2*n_i_*/*N_A_* and detectivity improvement by (2*n_i_*/*N_A_*)^1/2^.

In the π absorber with a cut-off wavelength of 6.5 μm at 300 K, the concentration of intrinsically generated holes drops to the acceptor doping level of 3 × 10^15^ cm^–3^ when the P^+^-π-N^+^ photodiode is reverse biased, which leads to an almost 4-fold decrease in the dark current. However, according to the above consideration, it is necessary to reduce the parasitic leakage mechanisms—tunneling and surface—to use the benefit of the suppression of Auger generation.

## Figures and Tables

**Figure 1 materials-17-02551-f001:**
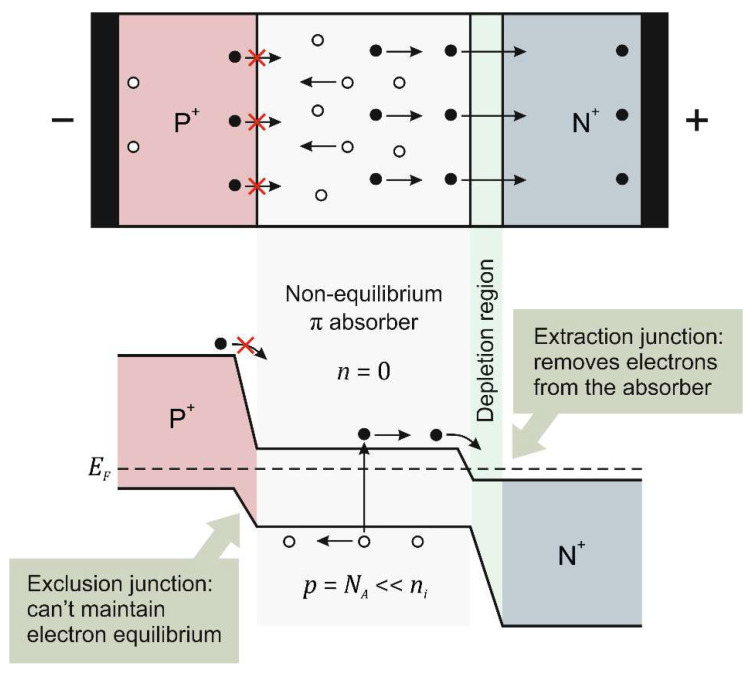
Principle operation of the non-equilibrium P^+^-π-N^+^ photodiode. Black dots indicate electrons, open dots indicate holes.

**Figure 2 materials-17-02551-f002:**
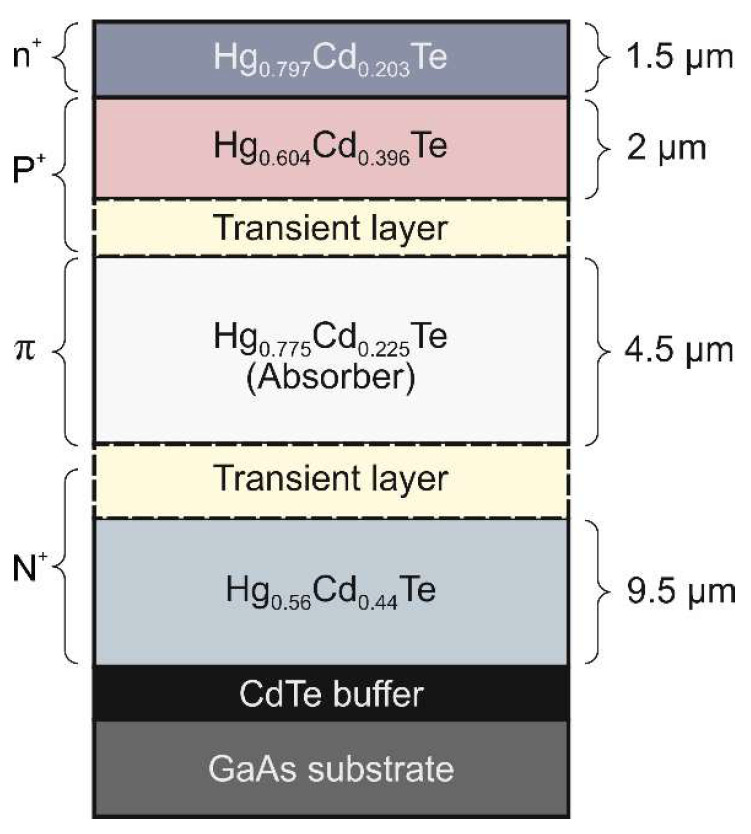
The cross-section of LWIR n^+^-P^+^-π-N^+^ HgCdTe photodiode.

**Figure 3 materials-17-02551-f003:**
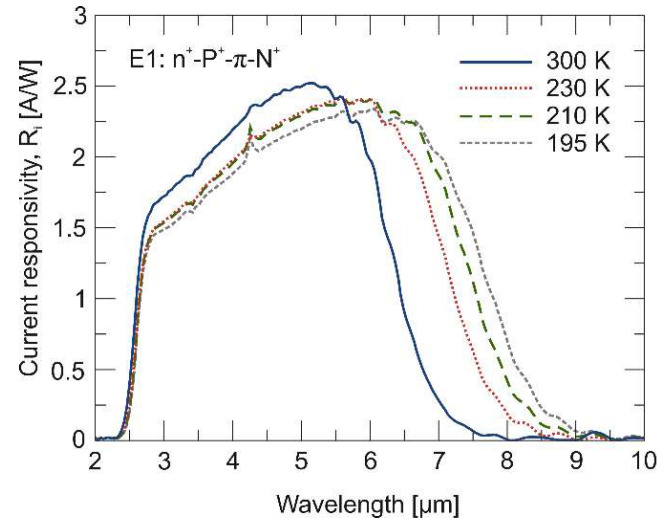
Measured spectral current responsivity (*R_i_*) of LWIR n^+^-P^+^-π-N^+^ HgCdTe photodiode.

**Figure 4 materials-17-02551-f004:**
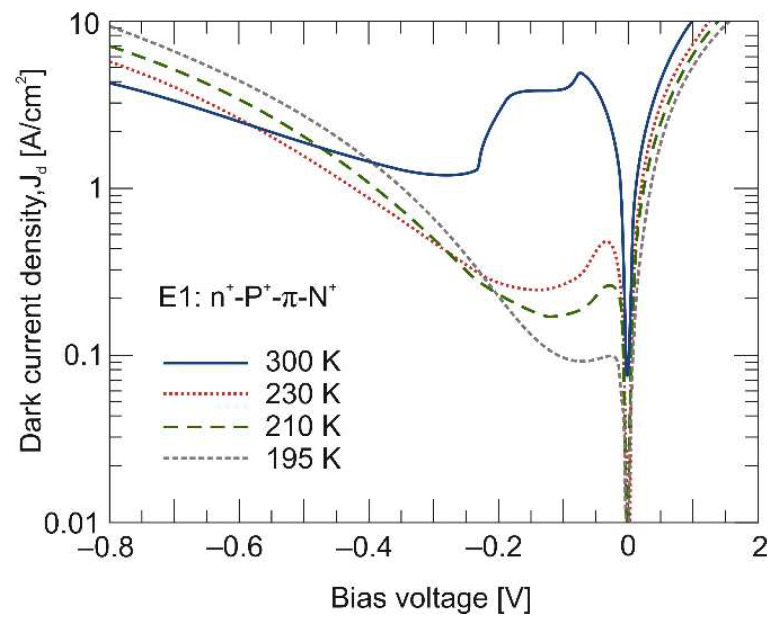
Measured dark current density–voltage (*J–V*) characteristics of LWIR n^+^-P^+^-π-N^+^ HgCdTe photodiode.

**Figure 5 materials-17-02551-f005:**
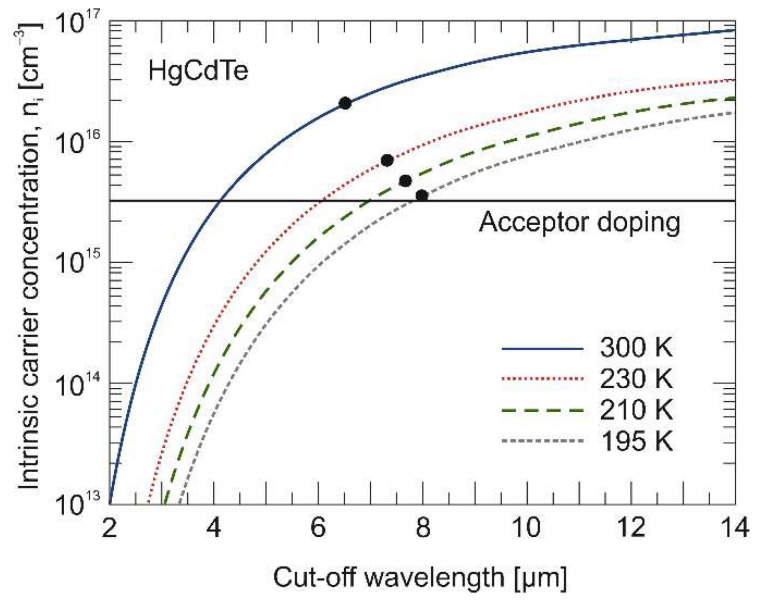
Calculated HgCdTe intrinsic carrier concentration (*ni*) versus cut-off wavelength. Black dots indicate cut-off wavelengths of the absorber at a given temperature. Black solid line marks an acceptor doping level of the absorber.

**Figure 6 materials-17-02551-f006:**
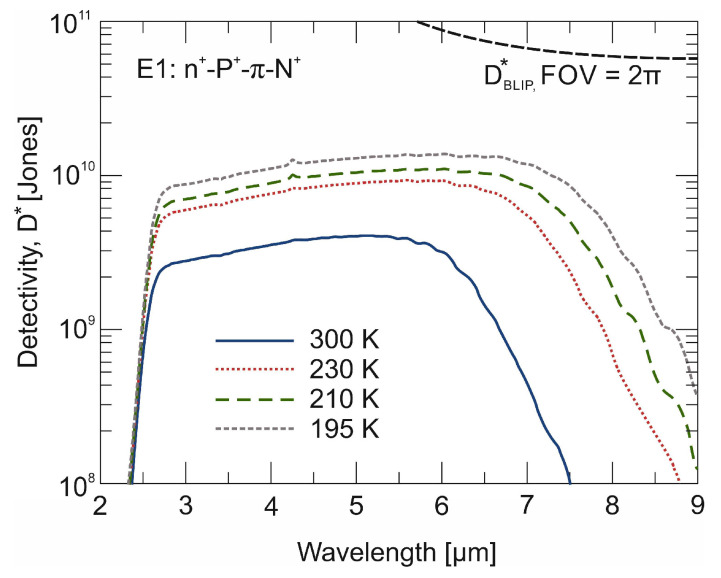
The normalized detectivity (*D**) of the LWIR n^+^-P^+^-π-N^+^ HgCdTe photodiode. The background-limited performance (BLIP) detectivity is calculated for the field of view (FOV) = 2π.

**Figure 7 materials-17-02551-f007:**
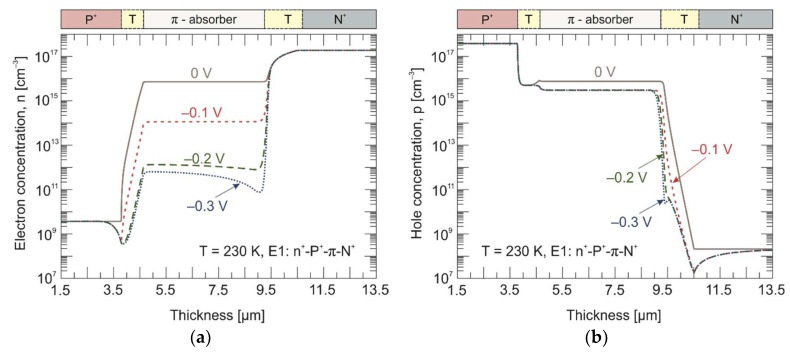
Calculated electron (**a**) and hole (**b**) concentration distribution across the P^+^-π-N^+^ HgCdTe heterostructure at 230 K. Solid and dotted/dashed lines represent equilibrium (0 V) and non-equilibrium (–0.1 V, –0.2 V, and –0.3 V) carrier concentrations, respectively.

**Figure 8 materials-17-02551-f008:**
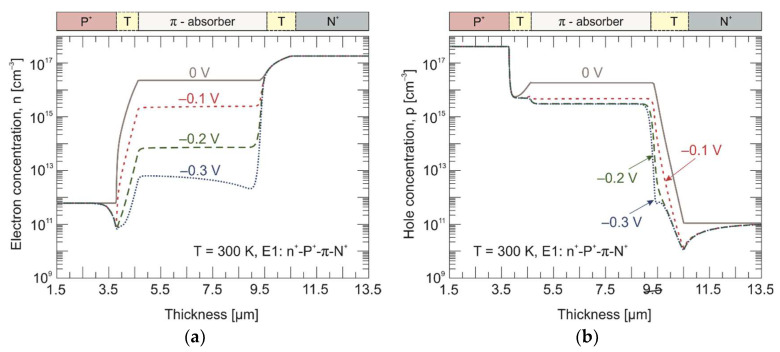
Calculated electron (**a**) and hole (**b**) concentration distribution across the P^+^-π-N^+^ HgCdTe heterostructure at 300 K. Solid and dotted/dashed lines represent equilibrium (0 V) and non-equilibrium (–0.1 V, –0.2 V, and –0.3 V) carrier concentrations, respectively.

**Figure 9 materials-17-02551-f009:**
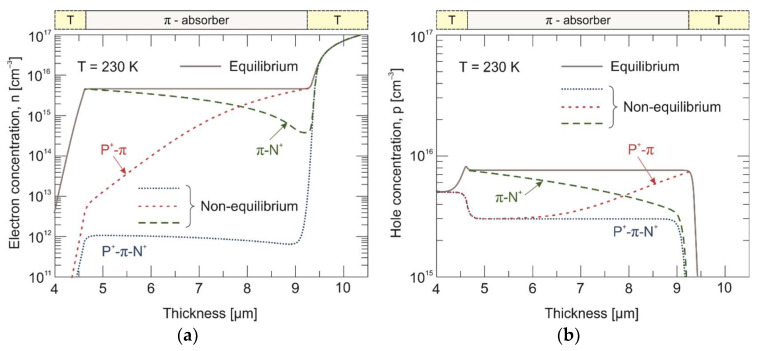
Calculated electron (**a**) and hole (**b**) concentration distribution across the P^+^-π-N^+^ HgCdTe heterostructure and separated n^+^-P^+^-π and π-N^+^ junctions at 230 K. Solid and dotted/dashed lines represent equilibrium (0 V) and non-equilibrium (–0.2 V) carrier concentrations, respectively.

**Figure 10 materials-17-02551-f010:**
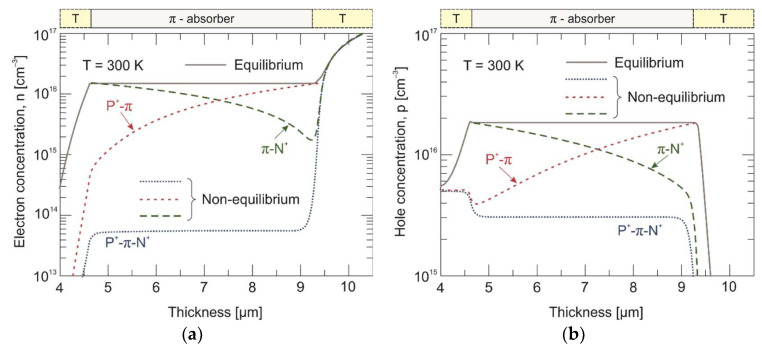
Calculated electron (**a**) and hole (**b**) concentration distribution across the P^+^-π-N^+^ HgCdTe heterostructure and separated n^+^-P^+^-π and π-N^+^ junctions at 300 K. Solid and dotted/dashed lines represent equilibrium (0 V) and non-equilibrium (–0.2 V) carrier concentrations, respectively.

**Figure 11 materials-17-02551-f011:**
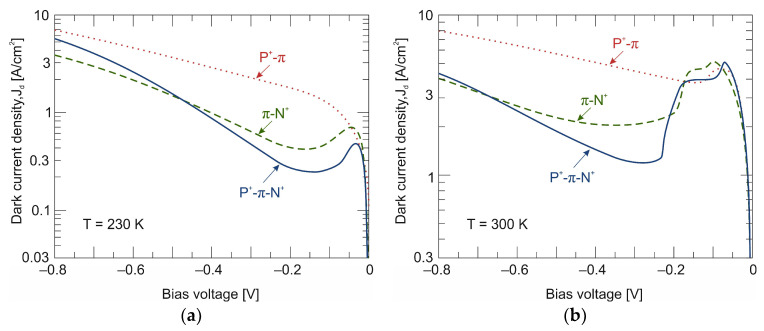
Measured dark current–voltage (*J–V*) characteristics of LWIR n^+^-P^+^-π-N^+^ HgCdTe photodiode and separated n^+^-P^+^-π and π-N^+^ junctions at 230 K (**a**) and 300 K (**b**).

## Data Availability

Data are contained within the article.

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
