# Peer review of "Non-Equilibrium Long-Wave Infrared HgCdTe Photodiodes: How the Exclusion and Extraction Junctions Work Separately"

_materials, 2024, doi:10.3390/ma17112551_

Round 1

Reviewer 1 Report

Comments and Suggestions for Authors

M. Kopytko et al. have mgiven the functioning of exclusion and extraction junctions separately in Non-equilibrium LWIR HgCdTe photodiodes. While the conceptual framework presented holds promise, several concerns need addressing before the paper can be considered for publication:

1.       The abstract requires revision to briefly highlight the paper's key findings or study outcomes. For instance, specifying whether three distinct measurements were conducted, or three different devices were evaluated would enhance clarity. Additionally, clarification is needed regarding the "two effects" mentioned in the statement "we focus on analyzing these two effects separately."

2.       It is essential to articulate how mitigating parasitic leakage mechanisms is imperative for attaining the intended objective.

3.       The paper should physical reasons behind the reduction in responsivity cutoff wavelength with increasing temperature, as depicted in Figure 3. Discuss more about theoretical relationship for HgCdTe or give reference  or providing relevant references would enrich the discussion.

4.       Including a detectivity plot would complement the analysis.

5.       The fluctuation in the diode's resistance from 0 to -0.2 requires explanation. A justification of the physical origins of dynamic resistance in such devices, along with temperature effects, is reasonable.

6.       Ensuring adequate referencing throughout the manuscript is imperative to bolster the credibility and contextualization of the presented findings.

Author Response

Thank you for your review. Answer in the attachment. 

Reviewer 2 Report

Comments and Suggestions for Authors

MCT detectors are commonly used for the detection of mid-infrared light. Kopytko et al. studied the mechanism of how the exclusion and extraction junctions help reduce the thermally generated dark current, therefore raising the working temperature of such detectors. 

They combined carrier density profile calculations and spectrally resolved leak current measurements on three MOCVD grown devices. The novelty of this work is that they studied the effects of exclusion and extraction separately and showed that exclusion has a more important effect. Their research design is sound and I recommend its publication. 

Author Response

Thank you for your review.

Reviewer 3 Report

Comments and Suggestions for Authors

In this work, two operation temperatures, 230 K and 300 K, for a HgCdTe IR detector were studied. The leakage current of this heterostructure photodiode in a P+ -π-N+ configuration was measured and compared to the simulation results by commercial APSYS program. From the simulation result, with the π absorber, at the cut-off wavelength of 6.5 μm and 300 K, the concentration of intrinsically generated holes drops to the acceptor doping level with reverse biased, resulting in a 4-fold decrease of the dark current was verified, which verified the feasibility operated at room temperature for this photodiode. This better performance of this configuration of photodiode was known for a long time. In this work, it just reverified by leak current measurement and by commercial program (APSYS) simulation. It is a fair work to be published, although it is not novel enough. In the followings are more correction should be done before the paper can be accepted.

In line 73 and 74, the “x” is never defined as Hg1-xCdxTe in the main text only in the figure. Please define the “x” in the main text.

In line 86, in the “H2O”, the “2” should be in subscript.

In Line 109: It is better to define “GR”, “SRH” in their full name at the first sight.

In line 128 and 130: the “cm2”, the “2” should be in superscripts.

In line 199: From the Fig. 8 and Fig. 9, it is very hard to see what author said: “At 300 K, the electron concentration drops by about an order of magnitude at both P+-π and π-N+junctions….”. It is difficult to compare when the 300 K and 230 K results were plotted in a separated figure. It recommends to plot together.

In line 216: the statement in authors’ claim as “Nevertheless, each of these effects (exclusion and extraction) is followed by the suppression of Auger generation…”. It is better to cite references for this argument and make a proof that why Auger generation is the dominated factor, which can be suppressed at 300 K.

In order to compare the dark current simulation in Fig 10(a) and Fig. 10(b), the vertical scale should be the same for easily recognition.

In line 238: “what leads to an…”, should the “what” changed to “which”?

Comments on the Quality of English Language

In line 238: “what leads to an…”, should the “what” changed to “which”?

Author Response

(The authors gave the same response as above.)

Round 2

Reviewer 1 Report

Comments and Suggestions for Authors

I am satisfied with the revision and paper can be accepted now. 

Reviewer 3 Report

Comments and Suggestions for Authors

It is acceptable.